# HGRN2: Gated Linear RNNs with State Expansion

[1]**Zhen Qin**[†], [2]**Songlin Yang**[†], [3]**Weixuan Sun**, [3]**Xuyang Shen**, [3]**Dong Li**, [3]**Weigao Sun**, [3]**Yiran Zhong**[*]

[1]TapTap [2]MIT CSAIL [3]OpenNLPLab, Shanghai AI Lab

⭘ https://github.com/OpenNLPLab/HGRN2

## Abstract

Hierarchically gated linear RNN (HGRN, Qin et al. 2023c) has demonstrated competitive training speed and performance in language modeling while offering efficient inference. However, the recurrent state size of HGRN remains relatively small, limiting its expressiveness. To address this issue, we introduce a simple outer product-based state expansion mechanism, which significantly enlarges the recurrent state size without introducing any additional parameters. This enhancement also provides a linear attention interpretation for HGRN2, enabling hardware-efficient training. Our extensive experiments verify the advantage of HGRN2 over HGRN consistently across different settings and comptetive to other recurrent models.

## 1 Introduction

Large language models (LLMs) have achieved significant empirical success in recent years. However, serving Transformer-based LLMs is costly due to the expensive KV cache management. Recurrent neural networks (RNNs), on the other hand, offer linear inference complexity with constant state size, making them ideal for serving. Consequently, there is substantial interest in studying **parallelizable** linear recurrent models, such as linear RNNs (Peng et al., 2023; Orvieto et al., 2023; Qin et al., 2023c; De et al., 2024), linear attention (Sun et al., 2023; Qin et al., 2023b; Yang et al., 2023; 2024; Arora et al., 2024), and state space models (Gu et al., 2022a; Smith et al., 2023; Gu & Dao, 2023; Dao & Gu, 2024).

RNNs have a fixed recurrent state size to encode all historical information. Therefore, it is important for RNNs to (i) utilize the fixed-sized states effectively and (ii) increase the recurrent state size to enhance memory capacity. Recent improvements in linear RNNs follow this approach, incorporating techniques such as data-dependent decays and state expansion.

Data-dependent decays (also known as forget gates) are crucial for RNNs (van der Westhuizen & Lasenby, 2018), allowing them to selectively retain useful information while erasing irrelevant information. This enables the fixed-size recurrent state to store only important information more efficiently. HGRN (Qin et al., 2023c) first emphasized the importance of data-dependent decays for linear RNNs. Many recent linear recurrent models, such as Mamba (Gu & Dao, 2023), Gated Linear Attention (GLA, Yang et al. 2023), Griffin (De et al., 2024), and RWKV-6 (Peng et al., 2024), also employ data-dependent decays.

However, HGRN did not increase the recurrent state size, which is greatly restricted by limited memory capacity. This limitation prevents it from achieving LLaMa-like (Touvron et al., 2023a;b) language modeling performance, as noted in Qin et al. (2024). Recent state-of-the-art linear recurrent models, such as Mamba, GLA, and RWKV-6, have addressed this issue by employing state-expansion techniques. These techniques significantly increase the recurrent state size and thereby enhance memory capacity, which has been shown to be crucial for language modeling performance and directly correlated with retrieval ability (Arora et al., 2024).

---

[*] Corresponding author. Email: zhongyiran@gmail.com. [†] Equal contributions.

In this work, we propose HGRN2, which aims to increase the recurrent state size for HGRN while retaining both parameter and training efficiency. We first explore structured matrices to expand the state size directly in a parameter-efficient manner. Empirically, we found that this approach improves language modeling performance but still encounters training inefficiencies, which limit the scaling of the recurrent state size. Inspired by linear attention, we then explore using a non-parametric outer product-based state expansion mechanism. This approach allows for efficient scaling of the recurrent state size during training without introducing additional parameters. Due to the matrix multiply form of linear attention, we can leverage the hardware-efficient linear attention training algorithm described in Yang et al. (2023); Qin et al. (2024) for large-scale experiments. As a result, HGRN2 can be regarded as an improved parameterization of GLA.

We extensively evaluate HGRN2 across various tasks, demonstrating that it consistently outperforms HGRN1 in multiple domains. In language modeling, we show HGRN2 to be highly competitive compared to other subquadratic efficient models.

## 2 Background

### 2.1 Gated linear RNN

Given input $\mathbf{x} \in \mathbb{R}^{N \times d}$, where the sequence length is $N$ and the model dimension is $d$, a minimalist gated linear recurrent layer (Martin & Cundy, 2018) transforms the input $\mathbf{x}$ into hidden states $\mathbf{h} \in \mathbb{R}^{N \times d}$ and the output $\mathbf{y} \in \mathbb{R}^{N \times d}$, as defined below:

$$
\begin{aligned}
\mathbf{g}_t &= \sigma\left(\mathbf{U}\mathbf{x}_t + \mathbf{b}_u\right), \\
\mathbf{i}_t &= \tau\left(\mathbf{V}\mathbf{x}_t + \mathbf{b}_v\right), \\
\mathbf{o}_t &= \sigma\left(\mathbf{W}\mathbf{x}_t + \mathbf{b}_w\right), \\
\mathbf{h}_t &= \mathbf{g}_t \odot \mathbf{h}_{t-1} + (1 - \mathbf{g}_t) \odot \mathbf{i}_t, \\
\mathbf{y}_t &= \mathbf{h}_t \odot \mathbf{o}_t,
\end{aligned}
\tag{1}
$$

where $\odot$ denotes element-wise product; $\sigma$ is the sigmoid function, and $\tau$ is a nonlinear activation function (we choose to use SiLU); $\mathbf{i}_t$ is the input vector; $\mathbf{g}_t$ and $\mathbf{o}_t$ are the forget gate and output gate, respectively. The input gate is tied to the forget gate as $1 - \mathbf{g}_t$, a common approach used in many gated RNNs such as GRU (Chung et al., 2014).

### 2.2 HGRN (Qin et al., 2023c)

Compared to Eq. 1, HGRN makes two adjustments: (i) complex-valued recurrence and (ii) forget gates with monotonically increased lower bound values from bottom layers to upper layers.

For (i), similar to the findings in Gu & Dao (2023) and De et al. (2024), we empirically found that complex-valued recurrence is not necessary, as shown in Table 1. The reason why HGRN found it useful is due to state expansion: the complex-valued recurrent state is twice the size of that in the real-valued recurrent state. If we directly expand the real-valued recurrent state size from $d$ to $2d$, the language modeling performance on the Wikitext-103 corpus is even better. Therefore, we only consider the real-valued recurrence thereafter.

Table 1: **Comparison of real HGRN and complex HGRN.** We found that real HGRN with twice the state size performs better than complex HGRN in Wiki103 language modeling.

| Method | State size | PPL(val) | PPL(test) | Params (M) |
|---|---|---|---|---|
| Complex HGRN1 | $2d$ | 24.14 | 24.82 | 46.25 |
| Real HGRN1 | $d$ | 25.34 | 26.12 | 46.24 |
| Real HGRN1 | $2d$ | 24.04 | 24.64 | 45.46 |

For (ii), suppose the total number of layers is $L$. HGRN introduces a data-independent learnable matrix $\Gamma \in \mathbb{R}^{L \times d}$, where $\Gamma_i$ represents the lowest values of the forget gate for the $i$-th layer at all time steps. HGRN argues that this lower bound should be monotonically increasing from bottom to top, encouraging the bottom layers to model short-term local

dependencies and the upper layers to model long-term dependencies. To enforce this monotonicity, HGRN uses the cumulative softmax operator `cumax` (Shen et al., 2018):

$$\beta := \mathtt{cumax}(\Gamma) = \mathtt{cumsum}(\mathtt{softmax}(\Gamma, \dim = 0), \dim = 0) \in \mathbb{R}^{L \times d}, \quad \beta^i = [\beta]_i \in \mathbb{R}^d.$$

To prevent the lower bound from reaching one in the highest layer, HGRN subtracts all $\beta$ values by $\beta^0$, making the lower bound for the first layer zero. After obtaining the lower bound values, the forget gate $\mathbf{g}_t$ learns the residuals instead, resulting in the new forget gate $\mathbf{f}_t$:

$$\begin{aligned}
\mathbf{f}_t^i &= \beta^i + (1 - \beta^i) \odot \mathbf{g}_t^i, \\
\mathbf{h}_t^i &= \mathbf{f}_t^i \odot \mathbf{h}_{t-1}^i + (1 - \mathbf{f}_t^i) \odot \mathbf{i}_t^i,
\end{aligned} \quad (2)$$

where the superscript indicates the layer index. This additive lower bound approach has been shown to mitigate the issue of saturated gates (Gu et al., 2020).

## 3 Method

### 3.1 Explorations of state expansion methods

The goal of this work is to scale the size of the HGRN recurrent state from $d$ to $nd$, where $n$ is the state expansion ratio. However, if we use the original parameterization in Eq. 1, the matrices $\mathbf{U}, \mathbf{V}, \mathbf{W}$ will have dimensions $d \times nd$, which becomes very parameter inefficient when $n$ is large. Ideally, the number of parameters should be around $d^2$, as in the original case for each projection. To achieve this, we first consider using structured matrices (e.g., low-rank matrices) to replace the dense projection matrix $\mathbb{R}^d \to \mathbb{R}^{nd}$, as described in Table 2.

Table 2: Parameter Efficient State Expansion (PESE) methods using Einstein Summation notation. **Blue** represents the input, **Black** represents data-independent weights, and **Red** represents the output. We list the Einstein Summation for low-rank (LR), group linear transformation (GLT), group linear transformation with interaction (GLTI), Khatri-Rao product (KRP), and Kronecker product (KP).

| Method | Equation | Parameter # |
|--------|----------|-------------|
| Naive | $d, \mathbf{d\ nd} \to nd$ | $nd^2$ |
| LR | $d, \mathbf{d\ r},\ \mathbf{r\ nd} \to nd$ | $dr(n+1) \approx d^2$ |
| GLT | $d = (n\ e) \to n\ e$ 
 $n\ e, \mathbf{n\ e\ d} \to n\ d$ | $d^2$ |
| GLTI | $d = (n\ e) \to n\ e$ 
 $n\ e, \mathbf{n\ e\ d},\ \mathbf{n\ n} \to nd$ | $d^2 + n^2$ |
| KRP | $d, \mathbf{n\ d} \to nd$ | $nd$ |
| KP | $d, \mathbf{d\ d},\ \mathbf{n} \to nd$ | $d^2 + n$ |

After obtaining the expanded $\mathbf{g}, \mathbf{i}, \mathbf{o}$, we feed them into element-wise gated linear recurrent layers as in Eq. 1 and Eq. 2, resulting in the output vector $\mathbf{y}_t \in \mathbb{R}^{n \times d}$. To project the expanded dimension back to the original dimension, we simply sum over the dimension corresponding to $n$.

The results are shown in Table 3. We found that state expansion generally improves performance, with the low-rank matrix performing the best among these candidates.

However, these methods face training inefficiency issues, as they require conducting element-wise linear recurrence in high dimensions (i.e., $nd$). Since these element-wise operations cannot leverage tensor cores (a fast matrix multiplication unit on GPUs), the dramatically increasing FLOPs and I/O costs significantly slow down training when $n$ is large. We notice that this is similar to the case in Mamba[1], which requires a relatively small expansion ratio (i.e., $n = 16$) and a custom I/O-efficient CUDA implementation to achieve a reasonably fast running speed.

In the next subsection, we explore an alternative strategy that does not replace the dense projection matrices with structured ones but instead changes the element-wise gating operations in Eq.1 to other matrix/vector operations similar to those used in linear attention. This approach allows for more efficient training.

Table 3: **PESE Ablation.** Ablation studies on various parameter-efficient methods, as described in Table 2. Each model was trained on 10 billion tokens from the Pile dataset.

| Method | n | PPL | Params (M) |
|--------|-----|------|------------|
| Xfmr | - | 5.16 | 380 |
| Xfmr++ | - | 4.62 | 386 |
| HGRN1 | 1 | 5.10 | 379 |
| LR | 4 | 4.76 | 385 |
| | 8 | 4.77 | 386 |
| GLT | 4 | 5.06 | 386 |
| GLTI | 4 | 4.83 | 386 |
| KRP | 4 | 5.08 | 386 |
| KP | 4 | 5.06 | 386 |
| HGRN2 | 4 | 4.79 | 385 |
| | 8 | 4.73 | 385 |
| | 128 | 4.62 | 385 |

## 3.2 HGRN2

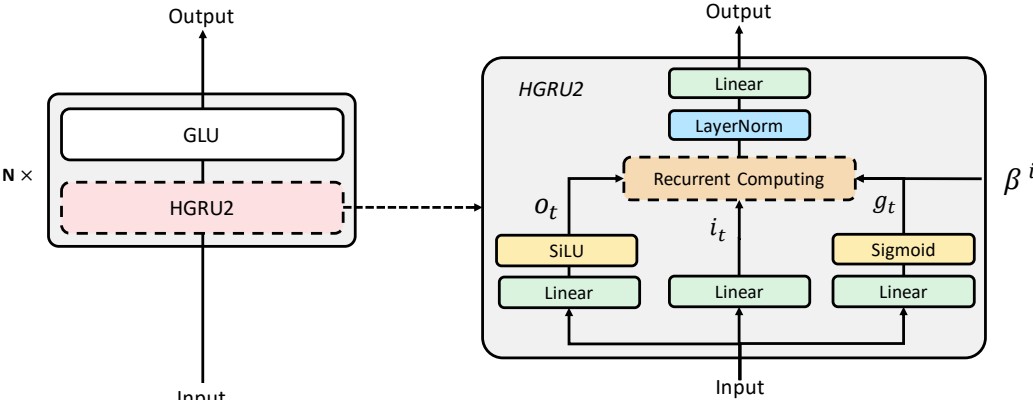

Figure 1: **Network Structure of HGRN2.** Each HGRN2 layer includes a token mixer layer, HGRU2, and a channel mixer, GLU. HGRU2 employs recurrent computation as described in Eq. 3, where $\mathbf{i}_t$ is the input state, $\mathbf{g}_t$ is the forget gate, $\mathbf{o}_t$ is the output gate, and $\beta^i$ is the lower bound for layer $i$.

The modification from HGRN1 to HGRN2 is simple yet effective. For the input gate, HGRN2 replaces the element-wise product with the outer product for state expansion. Consequently, $\mathbf{h}_t \in \mathbb{R}^{d \times d}$, and HGRN2 first diagonalizes the forget gate vector and uses the matrix dot product to update the hidden state. For the output gate, HGRN2 replaces the element-wise product with matrix-vector multiplication to project the expanded state back to the original dimension. The recurrent equation of HGRN2 is as follows:

$$\mathbf{h}_t = \mathbf{h}_{t-1} \cdot \mathrm{Diag}\{\mathbf{f}_t\} + \mathbf{i}_t \otimes (1 - \mathbf{f}_t) \in \mathbb{R}^{d \times d},$$
$$\mathbf{y}_t = \mathbf{h}_t \cdot \mathbf{o}_t \in \mathbb{R}^d,$$

(3)

where Diag denotes the diagonalization of vectors, $\cdot$ represents the matrix dot product, and $\otimes$ indicates the outer product.

---

[1]Though Mamba has an attention mechanism (Ali et al., 2024) similar to that in linear attention, the attention computation cannot be expressed as a matrix multiplication like linear attention, and thus does not facilitate tensor core-based GPU acceleration, as well acknowledged in Mamba2 (Dao & Gu, 2024).

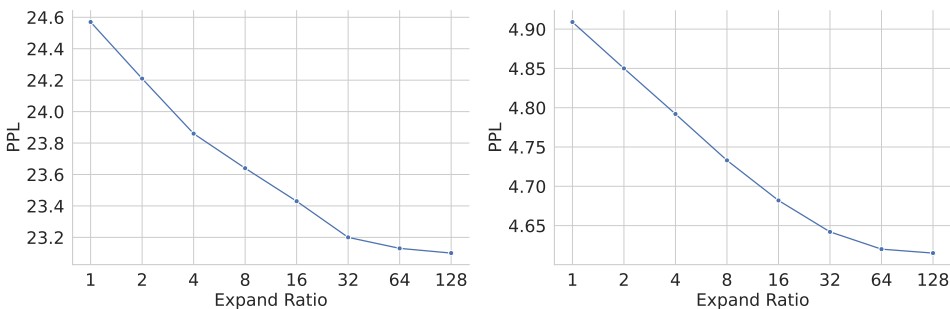

Figure 2: **Expand Ratio (Head Dimension) Ablation.** We tested the relationship between PPL (Perplexity) and the expand ratio on the Wikitext-103 (Merity et al., 2017) dataset (left) and a subset of the Pile (Gao et al., 2020) dataset (right).

**Multihead Variant.** The complexity of recurrence increases dramatically from $O(BNd)$ to $O(BNd^2)$ due to state expansion. To address this, we introduce a multihead variant of HGRN (similar to that in linear attention) such that the complexity is reduced to $O(BNd^2/H)$ for the number of heads $H$, effectively making the state size $d^2/H$, i.e., the expansion ratio $n = d_h = d/H$.[2] We conducted an ablation study on the expansion ratio (or head dimension) $n = \frac{d}{H}$, as shown in Figure 2. The results show that state expansion significantly improves language modeling performance. However, when the head dimension (i.e., state expansion ratio) exceeds 128, the performance gain diminishes. To balance computational cost and performance, we chose $d_h = 128$ for the main experiments.

**Comparison to GLA.** It is important to note that the recurrence form in HGRN2 is identical to that of GLA (Yang et al., 2023), except for the specific parameterization. We list the correspondences between the two parameterizations in Table 4. As shown, the output gate in HGRN2 corresponds to the query in GLA, while the output gate in GLA is omitted in HGRN2. The key vector in GLA corresponds to the input gate in HGRN2, which is tied to the forget gate, thereby saving parameters.

Table 4: The correspondence between HGRN2 and GLA is as follows.

| HGRN2 | GLA |
|:---:|:---:|
| **o** (output gate) | **q** (query vector) |
| $\mathbf{1} - \mathbf{f}$ (input gate) | **k** (key vector) |
| **i** (input vector) | **v** (value vector) |
| **f** (forget gate) | $\boldsymbol{\alpha}$ (forget gate) |
| − | **o** (output gate) |

**Hardware-Efficient Training.** Due to its computational structure's similarity to GLA, we can directly leverage their chunkwise algorithm and highly optimized kernels for hardware-efficient large-scale training. For more details, we refer readers to their paper.

**Concluding Remarks.** Although HGRN2 shares many similarities with GLA, we believe that HGRN2 offers a unique perspective distinct from linear attention, originating from the approach of gated linear RNNs. For instance, it may not be immediately clear from the perspective of linear attention why key vectors should be constrained within the range of (0, 1) or why the key vector and forget gate value should sum to one. However, these concepts become quite intuitive when starting from the gated linear RNN framework and exploring state expansion.

---

[2]See Bolya et al. (2022) for more detailed complexity analysis.

# 4 Experiments

## 4.1 MQAR

**Setting.** Multi-Query Associative Recall (MQAR) (Arora et al., 2023) is an enhanced version of the synthetic induction head dataset (Fu et al., 2023), designed to test the in-context associative recall ability of subquadratic models. Arora et al. (2023) found strong correlations between MQAR accuracy and language modeling performance. Our experimental setting strictly follows the original paper[3]. Our hyperparameter sweep included the following ranges: expansion ratio $\in \{64, 128\}$ and learning rate $\in \{1e-5, 5e-5, 1e-4, 5e-4, 1e-3, 5e-3, 1e-2\}$.

**Result.** As shown in Fig. 3, HGRN2 significantly outperforms HGRN1 across various model dimensions, demonstrating the benefits of state expansion in improving memory capacity and, consequently, in-context recall ability.

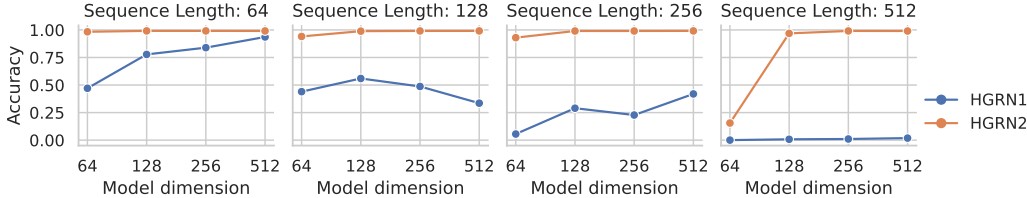

Figure 3: Results on MQAR, where the x-axis represents the model dimension and the y-axis represents accuracy. The task becomes more challenging as the sequence length increases. HGRN2 outperforms HGRN1 in all scenarios.

## 4.2 Language modeling

### 4.2.1 Wikitext-103

**Setting.** For the Wikitext-103 experiment, we followed the configuration of HGRN1 to validate the performance of 44M models against a wide range of subquadratic models: FLASH (Hua et al., 2022), 1+elu (Katharopoulos et al., 2020), Performer (Choromanski et al., 2021), cos-Former (Qin et al., 2022b), Syn(D), Syn(R) (Tay et al., 2021a), gMLP (Liu et al., 2021), S4 (Gu et al., 2022a), DSS (Gupta & Berant, 2022), RWKV-v4 (Peng et al., 2023), LRU (Orvieto et al., 2023), HGRN1 (Qin et al., 2023c), TNN (Qin et al., 2023a), and Mamba (Gu & Dao, 2023). All reported results are from our own runs under the same settings.

Table 5: **Results on Wikitext-103**.

| Model | PPL (val) | PPL (test) | Params (M) |
|---|---|---|---|
| Transformer | 24.40 | 24.78 | 44.65 |
| FLASH | 25.92 | 26.70 | 42.17 |
| 1+elu | 27.44 | 28.05 | 44.65 |
| Performer | 62.50 | 63.16 | 44.65 |
| cosFormer | 26.53 | 27.06 | 44.65 |
| Syn(D) | 31.31 | 32.43 | 46.75 |
| Syn(R) | 33.68 | 34.78 | 44.65 |
| gMLP | 28.08 | 29.13 | 47.83 |
| S4 | 38.34 | 39.66 | 45.69 |
| DSS | 39.39 | 41.07 | 45.73 |
| GSS | 29.61 | 30.74 | 43.84 |
| RWKV-4 | 24.31 | 25.07 | 46.23 |
| LRU | 29.86 | 31.12 | 46.24 |
| TNN | 23.98 | 24.67 | 48.68 |
| Mamba | 22.58 | 23.19 | 44.39 |
| HGRN1 | 24.14 | 24.82 | 46.25 |
| HGRN2 | 23.10 | 23.73 | 44.66 |

**Result.** Table 5 shows the results. HGRN2 clearly outperforms HGRN1 but slightly underperforms Mamba.

### 4.2.2 Slimpajama

We conducted language modeling experiments with 1.3B and 2.7B parameters on the Slimpajama dataset (Soboleva et al., 2023), using the FLASHLINEARATTENTION (Yang & Zhang, 2024) codebase for training. [4] The results, shown in Table 6, demonstrate that

---

[3]`https://github.com/HazyResearch/zoology`
[4]Model checkpoints are available at `https://huggingface.co/fla-hub`.

HGRN2 consistently outperforms other competitive linear recurrent models across three model scales. This suggests that HGRN2 provides a superior parameterization compared to GLA, as both models share an identical recurrent structure.

Table 6: Slimpajama language modeling results.

| | Lamb. ppl$\downarrow$ | Wiki. ppl$\downarrow$ | ARC$_e$ acc | ARC$_c$ acc | Hella. acc$_n$ | Lamb. acc$_n$ | PIQA acc | Wino. acc | Avg acc |
|---|---|---|---|---|---|---|---|---|---|
| *1.3B parameters with 100B training tokens* | | | | | | | | | |
| Transformer++ | 15.3 | 17.1 | 54.1 | 27.1 | 49.3 | 47.0 | 70.3 | 54.9 | 50.5 |
| Mamba | 16.5 | 18.2 | 57.3 | 26.6 | 48.1 | 43.4 | 69.5 | 53.7 | 49.8 |
| RetNet | 15.4 | 17.3 | 57.4 | 27.9 | 50.3 | 44.6 | 71.7 | 51.8 | 50.6 |
| GLA | 15.4 | 17.6 | 55.4 | 27.7 | 49.0 | 46.4 | 69.9 | 54.0 | 50.4 |
| HGRN2 | 11.8 | 16.9 | 58.1 | 28.1 | 51.8 | 49.4 | 71.4 | 52.3 | 51.9 |
| *2.7B parameters with 100B training tokens* | | | | | | | | | |
| Transformer++ | 10.7 | 15.2 | 59.8 | 27.5 | 54.2 | 52.3 | 72.7 | 56.2 | 53.8 |
| Mamba | 13.6 | 15.9 | 60.7 | 29.8 | 53.9 | 46.4 | 72.8 | 53.9 | 52.9 |
| RetNet | 11.9 | 15.8 | 59.6 | 28.1 | 54.0 | 49.6 | 72.3 | 53.8 | 52.9 |
| GLA | 12.4 | 15.5 | 59.2 | 29.9 | 54.0 | 50.4 | 71.7 | 55.7 | 53.5 |
| HGRN2 | 8.8 | 14.6 | 60.8 | 30.3 | 58.7 | 55.4 | 73.0 | 54.2 | 55.4 |

### 4.2.3 The Pile

We also conducted experiments on the Pile dataset. First, we trained 150M, 350M, and 1B HGRN1 and HGRN2 models for 100B tokens, and the results are shown in Table 7. We observe that HGRN2 consistently outperforms HGRN1.

Table 7: Comparison between HGRN1 and HGRN2 on Commonsense Reasoning Tasks.

| Model | Bn Params | Bn Tokens | PIQA | Hella. | Wino. | ARC-e | ARC-c | OBQA | AVG |
|---|---|---|---|---|---|---|---|---|---|
| HGRN1 | 0.15 | 100 | 65.02 | 33.33 | 50.20 | 46.68 | 23.81 | 28.60 | 41.27 |
| HGRN2 | 0.15 | 100 | 66.43 | 35.44 | 51.70 | 46.63 | 24.32 | 28.40 | 42.15 |
| HGRN1 | 0.35 | 100 | 66.70 | 38.12 | 51.70 | 49.20 | 25.26 | 30.60 | 43.60 |
| HGRN2 | 0.39 | 100 | 69.97 | 46.16 | 52.72 | 53.58 | 23.98 | 32.40 | 46.47 |
| HGRN1 | 1 | 100 | 70.89 | 48.02 | 51.62 | 55.64 | 27.90 | 31.60 | 47.61 |
| HGRN2 | 1 | 100 | 74.16 | 54.85 | 56.12 | 58.71 | 27.22 | 34.00 | 50.84 |

Next, we scaled the token horizon to 300B and trained strong baseline models, Mamba and LLaMA, under the same settings for comparison. We also compared them against several open-sourced language models, such as OPT (Zhang et al., 2022), Pythia (Biderman et al., 2023), BLOOM (Scao et al., 2022), and RWKV-4 (Peng et al., 2023). We found that HGRN2 performs competitively with Mamba, LLaMA, and other open-sourced LLMs.

To evaluate long-context abilities, we conducted tests on SCROLLs (Shaham et al., 2022) and found that HGRN2 exhibits better scaling behavior compared to Mamba, indicating stronger long-context capabilities, potentially due to its larger recurrent state size. However, we also observed that the 7B HGRN2 model is still not as strong as the LLaMA model, suggesting that the scaling behavior of linear models for long-context modeling remains an area for further study.

To test the retrieval ability of our trained 3B models, we ran the easy mode of the Needle in a Haystack Test. [5] LLaMA almost achieves perfect retrieval performance for evaluation

---

[5]In this mode (Shen, 2024; Shen et al., 2024), both the question and answer (QA pair) are embedded within a lengthy text, challenging the model to locate and respond to the query. This mode is particularly suitable for base models without instruction tuning. In contrast, the standard mode only places the answer within the long context, requiring the model to understand the question and find the relevant answer.

Table 8: Comparison between HGRN2 and other open-sourced language models, alongside strong baseline models (LLaMA and Mamba re-trained under the same settings), on Commonsense Reasoning Tasks. † indicates our own trained model.

| Model | Bn Params | Bn Token | PIQA | Hella. | Wino. | ARC-e | ARC-c | OBQA | AVG |
|---|---|---|---|---|---|---|---|---|---|
| OPT | 0.35 | 300 | 64.58 | 36.69 | 52.49 | 44.02 | 23.89 | 28.20 | 41.65 |
| Pythia | 0.40 | 300 | 67.08 | 40.52 | 53.59 | 51.81 | 24.15 | 29.40 | 44.43 |
| BLOOM | 0.56 | 350 | 64.09 | 46.97 | 52.80 | 47.35 | 29.38 | 28.20 | 42.23 |
| RWKV-4 | 0.43 | - | 67.52 | 39.00 | 51.14 | 52.86 | 25.17 | 32.40 | 45.00 |
| Llama† | 0.4 | 350 | 67.19 | 38.75 | 52.19 | 49.24 | 23.72 | 30.00 | 43.51 |
| Mamba† | 0.4 | 300 | 67.90 | 40.74 | 52.72 | 53.07 | 24.74 | 31.20 | 45.06 |
| HGRN2† | 0.4 | 300 | 67.74 | 40.32 | 51.78 | 54.21 | 24.83 | 31.20 | 45.01 |
| GPT-Neo | 1.3 | 300 | 71.11 | 48.93 | 54.93 | 56.19 | 25.85 | 33.60 | 48.44 |
| OPT | 1.3 | 300 | 71.71 | 53.70 | 59.35 | 57.24 | 29.69 | 33.20 | 50.82 |
| Pythia | 1.4 | 300 | 70.67 | 47.18 | 53.51 | 56.99 | 26.88 | 31.40 | 47.77 |
| BLOOM | 1.3 | 350 | 71.42 | 49.83 | 51.47 | 55.63 | 29.40 | 44.50 | 47.27 |
| RWKV-4 | 1.5 | - | 72.36 | 52.48 | 54.62 | 60.48 | 29.44 | 34.00 | 50.56 |
| Llama† | 1.0 | 300 | 69.97 | 47.04 | 52.72 | 57.07 | 26.18 | 32.60 | 47.93 |
| Mamba† | 1.0 | 300 | 71.27 | 50.15 | 56.35 | 58.71 | 29.27 | 31.20 | 49.45 |
| HGRN2† | 1.0 | 300 | 71.65 | 49.52 | 54.38 | 60.27 | 28.07 | 33.40 | 49.55 |
| OPT | 2.7 | 300 | 73.83 | 60.60 | 61.01 | 60.77 | 31.31 | 35.20 | 53.79 |
| Pythia | 2.8 | 300 | 74.10 | 59.31 | 59.91 | 64.14 | 33.02 | 35.60 | 54.35 |
| BLOOM | 3.0 | 350 | 70.57 | 54.53 | 58.49 | 59.43 | 30.38 | 32.20 | 50.77 |
| RWKV-4 | 3.0 | - | 72.42 | 58.75 | 57.30 | 62.92 | 35.15 | 36.20 | 53.79 |
| Llama† | 3.0 | 350 | 73.18 | 57.88 | 59.59 | 63.93 | 33.51 | 35.40 | 53.93 |
| Mamba† | 3.0 | 300 | 74.92 | 61.68 | 59.19 | 65.33 | 31.45 | 35.60 | 55.31 |
| HGRN2† | 3.0 | 300 | 74.10 | 61.48 | 58.64 | 65.61 | 34.47 | 35.60 | 54.98 |
| Llama† | 7.0 | 300 | 75.19 | 64.39 | 61.88 | 67.55 | 35.41 | 35.00 | 56.57 |
| HGRN2† | 7.0 | 300 | 76.50 | 66.96 | 61.40 | 69.02 | 36.86 | 38.00 | 58.12 |

Table 9: Performance Comparison on SCROLLS. R-1/2/L stand for parameter size, tokens, and rouge-1/rouge-2/rouge-l, respectively.

| Model | Params Bn | Token Bn | GovRep R-1/2/L | SumScr R-1/2/L | QMSum R-1/2/L | Qspr F1 | Nrtv F1 | QALT EM | CNLI EM | Avg ↑ |
|---|---|---|---|---|---|---|---|---|---|---|
| Llama | 0.4 | 300 | 8.2/3.5/6.2 | 11.3/1.6/8.7 | 10.7/2.1/9.4 | 17.8 | 15.4 | 28.0 | 13.9 | 10.5 |
| Mamba | 0.4 | 300 | 8.2/2.4/6.2 | 11.2/1.8/8.9 | 9.3/1.6/8.4 | 14.9 | 11.6 | 25.8 | 19.4 | 10.0 |
| HGRN2 | 0.4 | 300 | 15.3/3.5/10.9 | 7.4/0.8/6.2 | 8.3/1.2/7.4 | 12.4 | 10.9 | 26.4 | 31.5 | 10.9 |
| Llama | 1.0 | 300 | 12.9/3.1/9.4 | 9.5/0.8/7.7 | 10.9/2.2/9.4 | 22.8 | 16.0 | 28.4 | 9.9 | 11.0 |
| Mamba | 1.0 | 300 | 15.2/4.2/10.6 | 12.3/1.6/9.4 | 13.9/3.1/11.7 | 18.3 | 14.7 | 26.7 | 9.1 | 11.6 |
| HGRN2 | 1.0 | 300 | 14.9/4.2/10.5 | 11.4/1.4/9.2 | 10.9/2.3/9.7 | 16.2 | 15.1 | 27.8 | 10.6 | 11.1 |
| Llama | 3.0 | 300 | 11.2/4.9/8.1 | 11.9/1.9/9.3 | 16.1/4.3/12.9 | 28.6 | 20.8 | 30.4 | 20.2 | 13.9 |
| Mamba | 3.0 | 300 | 21.5/6.6/13.9 | 13.2/2.0/10.1 | 15.0/3.2/12.3 | 22.1 | 17.9 | 28.8 | 24.0 | 14.7 |
| HGRN2 | 3.0 | 300 | 21.7/6.6/14.1 | 14.6/2.1/10.8 | 12.5/2.7/10.6 | 25.4 | 18.8 | 28.9 | 31.9 | 15.4 |
| Llama | 7.0 | 300 | 17.4/7.3/11.4 | 12.9/1.8/10.0 | 14.6/3.7/11.8 | 32.4 | 22.3 | 33.8 | 10.0 | 14.6 |
| HGRN2 | 7.0 | 300 | 14.9/5.2/10.2 | 15.4/2.4/11.1 | 14.3/3.0/11.8 | 27.1 | 19.6 | 30.1 | 10.0 | 13.5 |

lengths no greater than the training length. As shown in Figure 4, HGRN2 and Mamba still face difficulties in retrieval tasks; however, HGRN2 outperforms Mamba due to its larger state size, enabled by linear attention-styled state expansion.

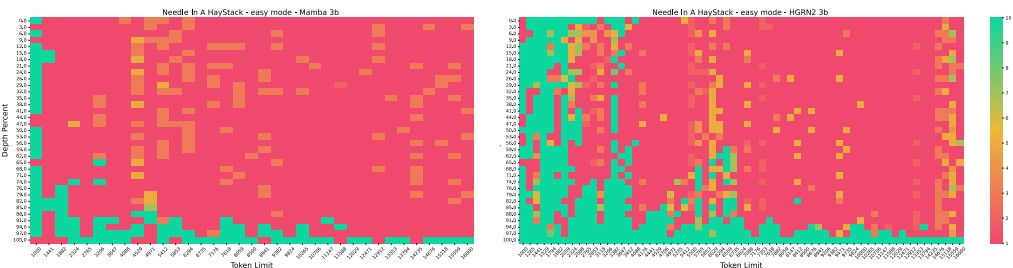

Figure 4: **Easy mode Needle in a Haystack Test on 3B models: Mamba (left) and HGRN2 (right)**. The evaluation context length is 16K, and the models were trained on a sequence length of 8K.

### 4.3 Long Range Arena

Table 10: Results on LRA. [†] indicates the results reported by Alonso et al. (2024).

| Model | ListOps | Text | Retrieval | Image | Pathfinder | Path-X | AVG |
|---|---|---|---|---|---|---|---|
| Transformer | 38.37 | 61.95 | 80.69 | 40.57 | 65.26 | - | 47.81 |
| cosFormer | 36.50 | 67.70 | 83.15 | 51.23 | 71.96 | - | 51.76 |
| FLASH | 38.70 | 64.10 | 86.10 | 47.40 | 70.25 | - | 51.09 |
| S4 | 59.60 | 86.82 | 90.90 | 88.65 | 94.20 | 96.35 | 86.09 |
| TNN | 61.04 | 87.90 | 90.97 | 88.24 | 93.00 | 96.10 | 86.21 |
| S5 | 62.15 | 89.31 | 91.40 | 88.00 | 95.33 | 98.56 | 87.46 |
| Mega | 63.14 | 90.43 | 91.25 | 90.44 | 96.01 | 97.98 | 88.21 |
| SGConv | 61.45 | 89.20 | 91.11 | 87.97 | 95.46 | 97.83 | 87.17 |
| LRU | 60.20 | 89.40 | 89.90 | 89.00 | 95.10 | 94.20 | 86.30 |
| Mamba[†] | 38.02 | 82.98 | 72.14 | 69.82 | 69.26 | 67.32 | 66.59 |
| Griffin[†] | 32.34 | 71.75 | 66.58 | 61.15 | 73.38 | 69.53 | 62.45 |
| HGRN1 | 59.95 | 88.14 | 94.23 | 88.69 | 92.92 | 97.50 | 86.91 |
| HGRN2 | 60.52 | 88.97 | 95.07 | 89.33 | 93.95 | 98.12 | 87.66 |

**Setting.** Long Range Arena (Tay et al., 2021b) is a benchmark designed to assess a model's ability to handle long-range dependencies. We used HGRN1's configuration and compared it with existing methods, as shown below.

**Result.** Table 10 shows the results. HGRN2 outperforms HGRN1, while Mamba and Griffin failed to achieve high accuracy on this benchmark.

### 4.4 Image Modeling

**Setting.** For the image classification task, we followed the configuration of HGRN1 and trained it on ImageNet-1k, comparing it with TNN and the vanilla transformer.

**Result.** Table 11 shows the results. HGRN2 outperforms HGRN1 with a similar parameter size, while also demonstrating an advantage over previous TNN (Qin et al., 2023a) and DeiT models (Touvron et al., 2021).

Table 11: Performances comparison of image classification on ImageNet-1k. HGRN2 performs favorably compared to competing methods with similar parameter sizes.

| | DeiT-Tiny | | DeiT-Small | |
|---|---|---|---|---|
| Model | Top-1 Acc | Params (M) | Top-1 Acc | Params (M) |
| DeiT | 72.20 | 5.7 | 79.90 | 22.0 |
| TNN | 72.29 | 6.4 | 79.20 | 23.4 |
| HGRN1 | 74.40 | 6.1 | 80.09 | 23.7 |
| HGRN2 | 75.39 | 6.1 | 80.12 | 23.8 |

## 5 Related work

**Linear recurrent models.** Linear recurrent models mainly include linear RNNs, state-space models, and linear attention. State-space models (SSMs) are gaining great attention since the seminal work S4 (Gu et al., 2022a) and its more efficient diagonalized version (Gu et al., 2022b). Despite excellent performance in the LRA benchmark, it has been shown to have inferior performance in language modeling. Gating mechanisms have been shown to be crucial in improving SSMs' language modeling performance (Mehta et al., 2023; Wang et al.,

2022; Gu & Dao, 2023). Gupta et al. (2022) build the connection between SSM and linear RNN. Orvieto et al. (2023) proposes a linear RNN layer (i.e., LRU) inspired by SSMs. Peng et al. (2023) successfully scale linear RNN models to billions of parameters for the first time.

For linear attention models, their language modeling performance has been underperforming softmax attention for a long time. Several improvements have been proposed to bridge the performance gap: (i) incorporating the forgetting mechanism (Peng et al., 2021; Schlag et al., 2021; Sun et al., 2023; Qin et al., 2023b; Yang et al., 2023; Peng et al., 2024), (ii) using local attention (Qin et al., 2022a; Zhang et al., 2023; Arora et al., 2024; Ren et al., 2024), (iii) using higher-order polynomial feature map (Arora et al., 2024; Kacham et al., 2023) to make the resulting attention distribution more sharp (Zhang et al., 2024), (iv) using more expressive yet efficient recurrent update rule (Schlag et al., 2021; Yang et al., 2024; Liu et al., 2024; Sun et al., 2024a).

**Gated linear recurrence.** Martin & Cundy (2018) first proposed a minimal gated linear recurrent layer and showed how to use the parallel scan algorithm to train linear RNNs in sequence-level parallel. Qin et al. (2023c) is largely based on this work with several adaptations and highlights the importance of data-dependent decay. De et al. (2024) build their model on LRU (Orvieto et al., 2023) and replace data-independent decays with data-dependent ones. They further use sliding-window attention to boost the performance. These models are limited in recurrent state size.

Gated recurrent models with matrix-valued recurrent state have been investigated in the literature of Neural Turing Machine (NTM Graves et al. 2014) and linear Transformer (Katharopoulos et al., 2020). In NTM, the number of memory slots can be regarded as the state expansion ratio discussed in this work. NTM also included data-dependent decays in the form of *erase vectors*. However, NTM is hard to parallelize and thus slow to train in practice. The linear transformer is known to have the recurrent form (Katharopoulos et al., 2020) and is known to be closely related to fast weight programming (FWP Schlag et al. 2021). Gated FWPs have been investigated since Schlag & Schmidhuber (2017); Zhang & Zhou (2017), and have recently been revisited in Peng et al. (2021); Mao (2022); Yang et al. (2023); Katsch (2023); Pramanik et al. (2023). In particular, Yang et al. (2023) proposed a hardware-efficient training algorithm for these types of models.

More recently, Mamba2 (Dao & Gu, 2024), xLSTM (Beck et al., 2024), and Gated Retention (Sun et al., 2024b) have shown that sharing data-dependent decays across different dimensions within the same head is effective. This approach improves efficiency over GLA because intra-chunk computations are more amenable to tensor core-based matrix multiplication acceleration, at the cost of sacrificing the fine-grainedness of decays. In GLA/HGRN2, each head dimension has its own decay rate, whereas in Mamba2/xLSTM/Gated Retention, all dimensions share the decay under a single head. It is an interesting question to study how much improvement fine-grained decay will bring.

# 6   Conclusion

In this work, we propose HGRN2, an enhancement of HGRN (Qin et al., 2023c) using an outer product-based state expansion mechanism inspired by linear attention, enabling efficient training. Experiments across multiple tasks validate the advantages of HGRN2 over HGRN1.

# Acknowledgement

We thank Yu Zhang for conducting some language modeling experiments and for the valuable discussions.

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

# A  Appendix

## A.1  Experiment Configurations

Table 12 provides detailed setups for both Auto-regressive Language Modeling (ALM) and ImageNet (IM) experiments, focusing on the WikiText-103 and ImageNet-1k datasets, respectively. The ALM experiments utilize Byte Pair Encoding (BPE) with a vocabulary size of 50,265 and a sequence length of 512, featuring a total batch size of 128 and 50,000 updates. The ImageNet experiments differentiate between 6 million and 23 million parameter models, with total batch sizes of 1024 and 2048, both running for 300 epochs but with differing warm-up periods. Optimization strategies vary between Adam for ALM and AdamW for IM, with specific learning rate schedulers and hyperparameters tailored to each model's scale. Additional configurations outline variations in model complexity, ranging from 0.15 to 2.9 million parameters, adjusting layers, hidden dimensions, and GPUs used, aiming to comprehensively explore model performance across scales and setups.

Table 12: **Comprehensive Configurations of the Model and Training Procedures for HGRN2 Experiments.** "Total batch size" means batch_per_gpu × update_freq × num_gpus; "ALM" stands for Autoregressive Language Model; "IM" stands for Image Modeling.

|  | ALM | IM(6M) | IM(23M) |
|---|---|---|---|
| Dataset | WikiText-103 | ImageNet-1k | ImageNet-1k |
| Tokenizer method | BPE | - | - |
| Src Vocab size | 50265 | - | - |
| Sequence length | 512 | - | - |
| Total batch size | 128 | 1024 | 2048 |
| Number of updates/epochs | 50k updates | 300 epochs | 300 epochs |
| Warmup steps/epochs | 4k steps | 20 epochs | 10 epochs |
| Peak learning rate | 5e-4 | 7.5e-4 | 7.5e-4 |
| Learning rate scheduler | Inverse sqrt | Cosine | Cosine |
| Optimizer | Adam | AdamW | AdamW |
| Adam $\epsilon$ | 1e-8 | 1e-8 | 1e-8 |
| Adam $(\beta_1, \beta_2)$ | (0.9, 0.98) | (0.9, 0.98) | (0.9, 0.98) |
| Weight decay | 0.1 | 0.05 | 0.1 |
| Gradient clipping | - | 5.0 | 5.0 |

Table 13: Model Configurations

| Params | Layers | Hidden Dim | Exp. Ratio | L. R. | Batch Size | SeqLen | GPUs |
|---|---|---|---|---|---|---|---|
| 0.15 | 15 | 768 | 128 | 3.00E-04 | 26 | 2048 | 8 |
| 0.385 | 26 | 1024 | 128 | 3.00E-04 | 15 | 2048 | 8 |
| 1 | 18 | 2048 | 128 | 3.00E-04 | 10 | 2048 | 16 |
| 2.9 | 36 | 2560 | 128 | 3.00E-04 | 36 | 2048 | 64 |

