# OpenReview forum: "HGRN2: Gated Linear RNNs with State Expansion"
_colmweb.org/COLM/2024/Conference — COLM_

### Official Review · Reviewer_9TbB · 2024-04-21

**Rating:** 6
**Confidence:** 2
**Ethics Flag:** 1

**Summary:**

The paper introduces HGRN2 which brings a simple but in practice significant improvements over the well performing HGRN model, an RNN based SOTA language model.

The paper is clear and easy to follow. The paper focuses on efficient state expansion to improve RNN model performance and the first part of section 3 motivates the state expansion.

Not only the approach is tested and compared with the most natural baseline HGRN1. The paper also test on various benchmarks (LGA, wiki, etc.) and compare with extensive number of models including Llama. The empirical results thus look convincing; the paper looks like a good contribution on RNN language models as alternatives to the main stream transformer based language models.

**Questions To Authors:**

I suppose that the performance in the paper refers to performance in test set, right? It will be good to make such points clear in the paper.

**Reasons To Accept:**

The paper follows recent trend to leverage RNN mechanisms to enable capturing long range dependencies, overcoming some inherent limitations of tranformer architectures. The paper is clear to communicate its contributions and situate well its core contributions within existing works from both recent RNN developments as well as transformers with respect to extending the range. The paper covers related works appropriately (e.g. relate to early state expansions techniques such as NTM).

The techniques proposed by the paper is very focused and can be implemented relatively easily. The simple changes and the significant gain of the proposal makes the techniques particularly attractive.

**Reasons To Reject:**

Although the paper has done arguably extensive comparisons over existing models, I find that the transfomer family models in the comparison are a bit outdated. For example, the transformer models in Table 6 and as well as the whole paper are around 2022-2023 while the field has made improvements to the architecture in tranformers and even Llama 2 is arguably a bit outdated (released July 2023). However, given that it is not the main focus of the current paper, this should be considered as a minor flaw of the paper.

The paper scales as large as 3B and it is still a relatively small model in today's standard. I understand the cost concern but some initial scaling law plot should be considered, since the question is how well the model scales is central for the actual adoption.

---

> ### Author Rebuttal · Authors · 2024-05-31
>
> Q1.  Compared transformer family models are a bit outdated.
>
> A1. Firstly, the network structures of Llama1/2/3 are almost identical, with the only difference being the tokens consumed. Therefore, the Transformer structure we provide remains state-of-the-art. We will add more results in future updates. As a clarification, the reason we did not include many updated models is because of the significant difference in token consumption. For example, Phi2 consumes 1 T tokens, while HGRN2 only consumes 100b, making the comparison unfair. As a supplement, we trained Llama and HGRN2 on the same dataset with 300B tokens sampled from the SlimPajama and Pile datasets, and it can be seen that HGRN2 performs better on most tasks. For the most result, please refer to Reviewer cvJL's A3. We also provide result on Scroll dataset:
>
> | Model | GovRep | SumScr | QMSUM | Qspr | QALT | CNLI |
> | --- | --- | --- | --- | --- | --- | --- |
> | HGRN2-385m | 15.78/3.72/11.26 | 7.13/0.76/6.03 | 8.03/1.28/6.99 | 12.39 | 26.46 | 31.53 |
> | Llama-385m | 7.51/2.21/5.92 | 9.96/1.26/7.76 | 6.32/0.78/5.71 | 17.48 | 27.95 | 13.5 |
> | HGRN2-1b | 20.26/5.28/13.66 | 11.83/1.72/9.11 | 9.65/1.74/8.82 | 19.82 | 27.61 | 11.38 |
> | Llama-1b | 7.53/2.13/6.19 | 9.34/1.33/7.31 | 7.82/1.17/6.4 | 18.99 | 27.85 | 26.9 |
> | HGRN2-3b | 21.89/6.7/14.2 | 15.02/2.15/10.83 | 12.48/2.61/10.46 | 25.2 | 28.81 | 31.92 |
> | Llama-3b | 9.77/2.95/7.27 | 10.39/1.49/8.1 | 7.72/1.46/6.27 | 28.8 | 30.58 | 20.15 |
>
> Q2. Provide some scaling law.
>
> A2. We sampled 300 billion tokens from the SlimPajama and Pile datasets and then retrained the HGRN2 model. We evaluated the model's performance on commonsense reasoning tasks and observed that the model's performance improves as more tokens are consumed.
>
> | Model | Tokens | PIQA | HS | WG | ARC-E | ARC-C | OBQA | AVG |
> | --- | --- | --- | --- | --- | --- | --- | --- | --- |
> | hgrn2_385m | 150 | 68.17 | 38.33 | 50.91 | 51.56 | 25.60 | 30.80 | 44.23 |
> | hgrn2_385m | 300 | 67.74 | 40.32 | 51.78 | 54.21 | 24.83 | 31.20 | 45.01 |
> | hgrn2_1b | 150 | 70.24 | 46.31 | 53.99 | 57.28 | 27.30 | 33.40 | 48.09 |
> | hgrn2_1b | 300 | 71.65 | 49.52 | 54.38 | 60.27 | 28.07 | 33.40 | 49.55 |
> | hgrn2_3b | 160 | 73.88 | 57.32 | 57.85 | 63.72 | 33.87 | 36.60 | 53.87 |
> | hgrn2_3b | 300 | 74.10 | 61.48 | 58.64 | 65.61 | 34.47 | 35.60 | 54.98 |
> | hgrn2_7b | 150 | 74.81 | 62.30 | 61.25 | 65.95 | 36.26 | 36.60 | 56.20 |
> | hgrn2_7b | 300 | 76.50 | 66.96 | 61.40 | 69.02 | 36.86 | 38.00 | 58.12 |

---

> > ### Comment · Reviewer_m3yg · 2024-06-03
> >
> > I'm not Reviewer 9TbB, but just wanted to ask a quick question about the long-context evals here. A lot of the benchmarks in scrolls are long context. When you performed the evaluations, did you truncate to the context length of the trained model like what they did in the [LongLLaMA](https://arxiv.org/abs/2309.16039) paper? Or did you just feed in the entire context without truncation?

---

> > > ### Author Response · Authors · 2024-06-04
> > > **Rebuttal by Authors**
> > >
> > > Thanks for your comments.  We feed in the entire context without truncation using the lm-eval-harness framework.

---

> > > > ### Comment · Reviewer_m3yg · 2024-06-04
> > > >
> > > > Hmm, in that case, then I find these SCROLLS results a bit strange. I believe the performance of Llama models would usually crash towards 0 if they are evaluated beyond the context length they were trained for (due to the positional encoding). Just to double-check: do you explicitly set the context length to be something very large like 100k? Otherwise, I believe lm-eval-harness might be doing truncation automatically to 2048 (or to the length that the model was trained for) (see [the max_length property here](https://github.com/EleutherAI/lm-evaluation-harness/blob/main/lm_eval/models/huggingface.py#L389-L400))
> > > >
> > > > (I also need to double-check again. Feel free to correct me if I'm wrong)

---

> > > > > ### Author Response · Authors · 2024-06-05
> > > > > **Response to Reviewer m3yg**
> > > > >
> > > > > As a supplement, we also tested the truncated version, and the results are as follows. It can be seen that the overall results are quite similar.
> > > > >
> > > > > | Model      | GovRep           | SumScr           | QMSUM            | Qspr  | QALT  | CNLI  |
> > > > > |------------|------------------|------------------|------------------|-------|-------|-------|
> > > > > | HGRN2-385m | 15.33/3.54/10.91 | 7.35/0.76/6.17   | 8.32/1.22/7.4    | 12.36 | 26.37 | 31.53 |
> > > > > | Llama-385m | 8.32/3.49/6.27   | 11.67/1.59/8.96  | 10.99/2.18/9.63  | 18.03 | 28.09 | 13.5  |
> > > > > | HGRN2-1b   | 20.44/5.32/13.78 | 11.25/1.71/8.83  | 10.45/1.91/9.33  | 19.86 | 27.56 | 11.38 |
> > > > > | Llama-1b   | 7.52/3.59/5.84   | 11.0/1.64/8.43   | 14.49/3.5/11.83  | 19.27 | 27.9  | 26.9  |
> > > > > | HGRN2-3b   | 21.7/6.62/14.09  | 14.55/2.13/10.79 | 12.48/2.69/10.58 | 25.41 | 28.86 | 31.92 |
> > > > > | Llama-3b   | 11.19/4.87/8.12  | 11.87/1.79/9.22  | 15.88/4.14/12.76 | 28.79 | 30.49 | 20.15 |

---

> > > > > > ### Comment · Reviewer_m3yg · 2024-06-06
> > > > > >
> > > > > > Makes sense, yeah, I believe this one is pretty consistent with what I would expect for a truncated version. Were you able to double check that the initial version was indeed using the full context?
> > > > > >
> > > > > > Thanks for providing these numbers!

---

> > > > > > > ### Author Response · Authors · 2024-06-06
> > > > > > > **Response to Reviewer m3yg**
> > > > > > >
> > > > > > > We rechecked the experiment and made sure that full context was used, and listed the actual sequence length below.
> > > > > > >
> > > > > > > | GovRep | SumScr | QMSUM | Qspr | QALT | CNLI |
> > > > > > > |--------|--------|-------|------|------|------|
> > > > > > > | 65k    | 23k    | 30k   | 21k  | 8.7k | 6.4K |

---

### Official Review · Reviewer_HShg · 2024-05-11

**Rating:** 6
**Confidence:** 3
**Ethics Flag:** 1

**Summary:**

HGRN2 presents extensions to HGRN by introducing a simple outer product-based state expansion mechanism, resulting in a larger recurrent state without introducing any additional parameters.  This corresponds to a linear attention interpretation of HGRN2 and hardware efficient training.  Mamba and GLA both use state expansion techniques.  HGRN has data dependent decays.  HGRN2 is evaluated on language modeling, image classification and long range arena.  The following sections describe background on the GRN and the HGRN.

The authors then explore parameter efficient state expansion methods, as shown in Table 2.  Section 3.2 then describes HGRN2, which does not use any of the approaches in Table 3.  They introduce a multi-headed HGRN as with linear attention.  Figure 2 shows better LM performance as the expansion ratio is expanded.  They compare HGRN to GLA and note all the similarities to the point that they can use their implementation.  They then perform various language modeling experiments, including WikiText and also long range arena.

There are some interesting experiments in this paper, and the updated HGRN could be an interesting contribution.  I have a few questions below and concerns about the experiments as described below.

* This paper would be much more impactful if it were planned to release the code for this?
* The Wikitext experiment needs more details.  For example, why wasn’t Mamba evaluated for perplexity?  Also, does perplexity hold for larger models (50M is relatively small).
* The approach seems pretty similar in to GLA in that they can share the same implementation and am concerned about differentiation from this work.  Can the authors expand on their description in the paper?
* Why wasn’t Mamba 2.8B in Table 6?  And perhaps Phi2 as well, which is more performant than the ones you put in the table.
* And for long range arena in Table 7, why not also evaluate on some newer benchmarks like Qasper?  But also, the table is deceiving, as the context length on which the model was trained isn’t listed in the table.  What context length was the transformer in the table trained on?  And HGRN2?  And did you try Rope extension for transformer models?  https://arxiv.org/html/2402.13753v1

Small things:
* Table 1 shows real HGRN1 with 1.5m more parameters, which seems like a lot and could be the reason it’s better than the complex version.  I don’t think this changes the result, but it’s sort of an unfair comparison.
* Would a larger expand ratio have (e.g., 256) have performed better?  Or were there diminishing returns?
* It would be nice if there was some other baseline (e.g., Transformer) in Fig. 3.
* Fig 4 (Right) is labeled HRGN instead of HGRN.

**Questions To Authors:**

See above in summary.

**Reasons To Accept:**

The paper presents incremental extensions to HGRN1 and some performance gains as a result over HGRN1.

**Reasons To Reject:**

* The results aren't well-described and at times inconsistent (see above).
* The approach seems to not differentiate much from GLA after performing the new state expansion, which makes the contributions somewhat limited.

---

> ### Author Rebuttal · Authors · 2024-05-31
>
> Q1. Are there plans to open-source the code?
>
> A1. Yes, we plan to open-source the code very soon.
>
> Q2. More about Wikitext experiment and larger model result.
>
> A2. We have updated the results of Mamba on the WikiText-103 dataset, as shown in the table below; additionally, we tested Mamba on 10 billion tokens in a subset of the Pile and observed that Hgrn2 outperforms Mamba.
>
> Wikitext-result:
> | Method | PPL(val) | PPL(test) | Params |
> | --- | --- | --- | --- |
> | HGRN2 | 23.1 | 23.73 | 44.66 |
> | Mamba | 22.58 | 23.19 | 44.39 |
>
> 10b token result, the same as figure 4.
> | Method | PPL |
> | --- | --- |
> | Llama-3b | 7.129 |
> | HGRN2-3b | 6.989 |
> | Mamba-3b | 7.056 |
>
> Q3. Relation to GLA and the novelty of HGRN2.
>
> A3. For most part, please refer to Reviewer cvJL' A1. In comparison with GLA, it is noteworthy that the parameters of keys and decay rates in linear attention can be shared. To the best of our knowledge, this has not been proposed previously in the literature. Our model also eliminates the output gate of linear attention, making the overall architecture more concise.
>
> Q4. More about table 6.
>
> A4. In future updates, we will add more results. For the reason, please refer to Reviewer m3yg'A1.
>
> Q5. More about Table 7.
>
> A5. The transformer results for the table are sourced from [1]. All models are trained from scratch, so there is no issue of extrapolation. In the future, we will also denote the sequence length for each task.
>
> Q6. More long context result.
>
> A6. Please refer to Reviewer 9TbB's A1.
>
> Q7. Update Table 1.
>
> A7. We have updated Table 1 to provide a version with fewer parameters, expand 2. As you can see, the performance is still better.
>
> | Method | State size | PPL(val) | PPL(test) | Params |
> | --- | --- | --- | --- | --- |
> | HGRN1 | d | 24.14 | 24.82 | 46.25 |
> | Real HGRN1 | 2d | 24.04 | 24.64 | 45.46 |
>
> Q8. Larger expand ratio.
>
> A8. We provide the results of expand 256 in the table below. As seen, the performance is better but the improvement is not significant. As a compromise, we chose an expand ratio of 128 for lm.
>
> | expand ratio | PPL(val) | PPL(test) |
> | --- | --- | --- |
> | 128 | 23.1 | 23.73 |
> | 256 | 23.41 | 23.79 |
>
> Q9. Transformer Baseline for MQAR.
>
> A9. We will provide it in future versions.
>
> Q10. Fig 4 typo.
>
> A10. We will fix this issue in the revised paper.
>
> Citations:
> [1] Yi Tay, et. Long Range Arena : A Benchmark for Efficient Transformers . In *International Conference on Learning Representations*.

---

> > ### Comment · Reviewer_HShg · 2024-06-05
> > **Response**
> >
> > * For Mamba, it appears to be better than HGRN2 on perplexity, is that right?  If you agree, then and put the Mamba result in the paper, that would be helpful.
> > * For table 6, what happens with more tokens (it looks like you only compared at 100B token size for Mamba).  Do the results hold at larger token sizes (e.g., 300B)?
> > * For table 7, please add context length to the paper.  Context length does matter, as linear models will do well with short context, but potentially not long context, and could be one of the advantages of approaches like these.
> > * For 256 expand ratio, why do you think there is diminishing returns?
> >
> > If you can address these, I would consider raising my score.

---

> > ### Author Response · Authors · 2024-06-05
> > **Response to Reviewer HShg**
> >
> > Q1. About Mamba result.
> >
> > A1. According to our experimental results, on Wikitext-103, mamba outperforms Hgrn2; as the model size increases (to 3b scale), Hgrn2 outperforms mamba. We will add the results of mamba in future versions.
> >
> > Q2. More about table 6.
> >
> > A2. For the fairest comparison, we re trained llama and hgrn2 on the same dataset for 300b tokens, as shown in the follow table. Due to time constraints, we did not train mamba. However, based on the existing results, Hgrn2 outperformed Llama. Training of mamba will be conducted in future.
> >
> > | Model | PIQA | HS | WG | ARC-E | ARC-C | OBQA | AVG |
> > | --- | --- | --- | --- | --- | --- | --- | --- |
> > | HGRN2-385m | 68.01 | 40.37 | 52.25 | 53.66 | 24.32 | 31.20 | 44.97 |
> > | Llama-385m | 67.19 | 38.75 | 52.17 | 49.24 | 23.72 | 30.00 | 43.51 |
> > | HGRN2-1b | 71.60 | 49.45 | 53.91 | 60.40 | 28.24 | 33.20 | 49.47 |
> > | Llama-1b | 69.97 | 47.04 | 52.72 | 57.07 | 28.16 | 32.60 | 47.93 |
> > | HGRN2-3b | 74.37 | 61.49 | 58.41 | 65.49 | 34.56 | 36.00 | 55.05 |
> > | Llama-3b | 73.18 | 57.88 | 59.59 | 63.93 | 31.40 | 34.00 | 53.33 |
> > | HGRN2-7b | 76.77 | 66.81 | 60.93 | 69.15 | 36.43 | 38.00 | 58.02 |
> > | Llama-7b | 75.19 | 64.39 | 61.88 | 67.55 | 35.41 | 35.00 | 56.57 |
> >
> > Q3. Add context length to table 7.
> >
> > A3. In the upcoming versions, we will add the sequence length in the table. For now, we provide the sequence length for each task.
> >
> > | Data | ListOps | Text | Retrieval | Image | Pathfinder | Path-X |
> > | --- | --- | --- | --- | --- | --- | --- |
> > | Seqlen | 2k | 4k | 4k | 1k | 1k | 16k |
> >
> > Q4. The reason for diminishing returns for the 256 expand ratio
> >
> > A4. This may be related to the sequence length; our current sequence length is 512, and at this sequence length, 128 is likely sufficient for Wikitext-103.

---

> > > ### Author Response · Authors · 2024-06-06
> > > **Looking forward to your reply**
> > >
> > > Dear Reviewer HShg,
> > >
> > > Could you please let us know if our response has addressed your concerns? If you have any further questions, please feel free to raise them at any time.

---

> > > ### Comment · Reviewer_HShg · 2024-06-06
> > > **Response**
> > >
> > > Under the assumption that the above are added to the final version of the paper, including ideally a Mamba result, I will raise my score.

---

### Official Review · Reviewer_cvJL · 2024-05-11

**Rating:** 6
**Confidence:** 4
**Ethics Flag:** 1

**Summary:**

The paper presents a modification to an existing recurrent neural network architecture called hierarchically gated linear RNN. The goal is to improve the expressiveness that is limited by its small state size. This is a niche topic, but given the increased interest to RNN-like alternatives for Transformers, standard RNN architectures are of interest to the community.

Specifically, they achieve the state size expansion by employing outer product operation on the hidden state and then sum the expanded representations to a single vector. In terms of originality, this technique is not new as it has been previously used in several linear attention works although it is interesting to see it applied in this standard RNN architecture.

Evaluation covers a number of tasks including multi-query associative recall, language modeling, commonsense reasoning, and image classification. The main baseline is the hierarchical gated RNN and other efficient architectures such as Mamba, RWKV, Griffin, and others. Even though coverage is fair, the results would have been more convincing if it covered more on downstream generative tasks.

Overall, it's a focused contribution with strong empirical focus but it has a limited scope and it lacks methodological novelty.

**Questions To Authors:**

- I was wondering if you can report scores on some of the general benchmarks such as MMLU, BBH or DROP that large language models are typically being evaluated on. This would help show that the method can generalize more broadly.

**Reasons To Accept:**

- Straight-forward, clear exposition and a simple method that address the capacity issue in hierarchically gated RNNs. It can also be applied to any gated-recurrent network.
- Addresses a problem that is not typical in the present literature; this attempt is forward looking and led to interesting findings. For instance, the improved gated RNN actually outperforms stronger efficient architectures based on Transformers.
- The claims made in the introduction are supported well by empirical evidence; the main target of comparison is the hierarchical gated RNN.

**Reasons To Reject:**

- The proposed method for addressing the capacity issue is from prior work and its methodological novelty is limited.
- The evaluation focuses mainly on improving hierarchical gated RNN only that is restrictive for the scope of the study.

---

> ### Author Rebuttal · Authors · 2024-05-31
>
> Q1. The novelty of this work
>
> A1. The primary contribution of this work is to recognize and emphasize the significance of state expansion and to offer  a simple yet elegant solution for expanding the recurrent state. In comparison to complex network structure modifications, we believe that our adjustment to HGRN is more foundational and addresses the core weakness of HGRN. Additionally, we provide analysis that supports our conclusion and offer comprehensive evaluations of state expansion strategies.  In terms of structure modification, our model leverateges parameter sharing for keys and decay rates in linear attention and eliminates the output gate of linear attention, making the overall architecture concise.
>
> Q2. Restrictive for the scope of the study.
>
> A2. Thank you for your question. In an additional experiment, we want to show that state expansion works equally well for general linear RNNs. We compared the perplexity (ppl) at different expansion ratios on the WikiText-103 dataset for a basic linear RNN(i.e. HGRN2 without the lower bound). We found that as the expansion ratio increases, the perplexity decreases.
>
> | Expand ratio | PPL(val) | PPL(test) |
> | --- | --- | --- |
> | 1 | 25.17 | 26.01 |
> | 2 | 24.56 | 25.41 |
> | 4 | 24.27 | 25.05 |
> | 8 | 23.86 | 24.51 |
> | 16 | 23.69 | 24.28 |
> | 32 | 23.28 | 23.93 |
> | 64 | 23.1 | 23.73 |
> | 128 | 23.21 | 23.76 |
>
> Q3. More evaluation result.
>
> A3. For a fair compare, we trained Llama and HGRN2 on the same dataset with 300B tokens sampled from the SlimPajama and Pile datasets, and it can be seen that HGRN2 performs better on most tasks.
> | Model | PIQA | HS | WG | ARC-E | ARC-C | OBQA | CSR-AVG | DROP | MMLU |
> | --- | --- | --- | --- | --- | --- | --- | --- | --- | --- |
> | HGRN2-385m | 68.01 | 40.37 | 52.25 | 53.66 | 24.32 | 31.20 | 44.97 | 4.16 | 25.07 |
> | Llama-385m | 67.19 | 38.75 | 52.17 | 49.24 | 23.72 | 30.00 | 43.51 | 3.95 | 27.08 |
> | HGRN2-1b | 71.60 | 49.45 | 53.91 | 60.40 | 28.24 | 33.20 | 49.47 | 3.42 | 25.47 |
> | Llama-1b | 69.97 | 47.04 | 52.72 | 57.07 | 28.16 | 32.60 | 47.93 | 5.88 | 24.9 |
> | HGRN2-3b | 74.37 | 61.49 | 58.41 | 65.49 | 34.56 | 36.00 | 55.05 | 5.20 | 25.33 |
> | Llama-3b | 73.18 | 57.88 | 59.59 | 63.93 | 31.40 | 34.00 | 53.33 | 3.92 | 27.52 |
> | HGRN2-7b | 76.77 | 66.81 | 60.93 | 69.15 | 36.43 | 38.00 | 58.02 | 5.67 | 32.99 |
> | Llama-7b | 75.19 | 64.39 | 61.88 | 67.55 | 35.41 | 35.00 | 56.57 | 4.19 | 32.2 |

---

> > ### Comment · Reviewer_cvJL · 2024-06-04
> > **Response to Rebuttal**
> >
> > Thanks for the response and the additional efforts.
> >
> > - Besides the technical component of state expansion itself, I agree that the study has novel aspects and I'd encourage the authors to emphasize them in the final version.
> >
> > - To support the claim regarding effectiveness and applicability in RNNs more broadly, I'd suggest providing empirical evidence different architectures such as LSTM, GRU, standard RNN.
> >
> > - Regarding the downstream performance, my concern has been addressed. It's encouraging that the improvement holds as the model size increases.

---

### Official Review · Reviewer_m3yg · 2024-05-11

**Rating:** 6
**Confidence:** 3
**Ethics Flag:** 1

**Summary:**

The paper proposes HGRN2 (Hierarchically Gated Linear RNN2), which is an improvement over the previous [HGRN](https://arxiv.org/abs/2311.04823). The main changes of HGRN2 are state expansion of the recurrent state from d to nd, which is done through some clever matrix multiplication and diagonalization tricks (exact details in Sec 3.2). The paper evaluates on various tasks such as Multi-Query Associative Recall (MQAR), language modeling on WikiText (perplexity eval), downstream common sense evals, and long-range arena. It shows that HGRN2 outperforms HGRN1 on most tasks, and it also performs a number of other models such as Mamba, OPT, Pythia, etc.

Also, a quick timeline here, in case it helps other reviewers/ACs who are wondering why the paper did not discuss baseline X:
- COLM deadline: March 29
- [Jamba](https://arxiv.org/abs/2403.19887) release: March 28 (probably too late to include in the paper)
- [Infini-attention](https://arxiv.org/abs/2404.07143) release: April 10
- [RecurrentGemma](https://arxiv.org/abs/2404.07839) release: April 11

**Questions To Authors:**

- Why are your Mamba results in Table 6 different form the results reported in the [Mamba](https://arxiv.org/pdf/2312.00752) paper?
   - Also, why is Mamba there for the 1B models but not there for the 3B models? The original paper/release had both 1B and 3B models.



[Note: Increased my score after rebuttals (see discussion below)]

**Reasons To Accept:**

- I feel like linear models are generally becoming quite popular these days, so this paper is very much aligned with the trend of linear attention research in 2024.
- Extensive evaluation setup, covering a wide range of tasks and settings. Good comparisons with other models. I especially liked Tables 6 and 7, where a lot of models were compared against, including both transformer models and other linear models/state-space models.
- The paper evaluated on actual natural language tasks! I'm very happy to see this, since I've seen a lot of papers in this area that report only validation loss/perplexity, so it's nice to see this paper actually cover downstream natural language tasks and commonsense tasks.

**Reasons To Reject:**

- Incremental improvements over the previous work HGRN (This is not to say that it's not good work; I just feel like it might be more suitable for a blog/technical report rather than a full conference submission).
- The transformer models that the paper compares with are weak -- OPT, BLOOM, and Pythia are known to be pretty bad models. For smaller sizes, there are stronger models such as Gemma-2B, StableLM2-1.6B, and phi2. I suspect that if the paper compares with these models, then these will probably perform better than HGRN2, since the weak models like BLOOM already perform quite close to it.
   - As a consequence of above: If the models fail to beat already existing good transformer models, then what's the whole point to begin with?
- I'm also not very convinced about long-context evals. I feel like long range arena is somewhat outdated and that a lot of the tasks feel like toy tasks and not very related to natural-language. These days there are a lot of newer evals such as needle-in-the-haystack style evals or natural language evals like [SCROLLS](https://arxiv.org/abs/2201.03533).

---

> ### Author Rebuttal · Authors · 2024-05-31
>
> Q1. Incremental improvements over the previous work HGRN.
>
> A1. The primary contribution of this work is to recognize and emphasize the significance of state expansion and to offer  a simple yet elegant solution for expanding the recurrent state. In comparison to complex network structure modifications, we believe that our adjustment to HGRN is more foundational and addresses the core weakness of HGRN. Additionally, we provide analysis that supports our conclusion and offer comprehensive evaluations of state expansion strategies. Given these factors, we are confident that HGRN2 is a solid work for submission to a full conference.
>
> Q2. The transformer models that the paper compares with are weak.
>
> A2. The performance of a model depends on two factors: the model's capacity and token consumption. To make a fair comparison based on model capacity, the number of tokens consumed should be similar. We have provided these baselines because their token consumption (around 300B) is relatively close to our token consumption (around 100B). In contrast, updated transformer models such as Gemma-2B, StableLM2-1.6B, and phi2 consume around or more than 1T tokens, which we believe would make the comparison highly unfair. For a fair comparison result, please refer to Reviewer cvJL's A3.
>
> Q3. More long-context evals.
>
> A3. Please refer to Reviewer 9TbB's A1.
>
> Q4. Why are your Mamba results in Table 6 different form the results reported in the Mamba paper?
>
> A4. We are using the same setup as described in [1] and training our models with 100 billion tokens. To ensure a fair comparison, we are referencing the Mamba results from [1]. The original Mamba results, as outlined in the Mamba paper, are derived from a model trained with 300 billion tokens.
>
> Q5. Why is Mamba there for the 1B models but not there for the 3B models?
>
> A5. Our model only used 100 billion tokens, while the original Mamba 3B used 300 billion tokens, making the comparison unfair. For a fair comparison, we trained on 10 billion tokens from the same dataset. The results are presented in the table below, showing that HGRN2 has a lower perplexity (ppl) at 3 billion.
>
> | Method | PPL |
> | --- | --- |
> | Llama-3b | 7.129 |
> | HGRN2-3b | 6.989 |
> | Mamba-3b | 7.056 |
>
> Citation.
> 1. Yang et. Gated Linear Attention Transformers with Hardware-Efficient Training.

---

> > ### Comment · Reviewer_m3yg · 2024-06-03
> >
> > Thanks for your responses. I think at the core of my concerns is really just trying to make sure that the method is something that's indeed competitive with other models out there.
> >
> > Some of my concerns have been addressed, but I still don't feel entirely convinced.
> > - Figure 4 is good, but not enough. I think loss is a good indicator *if* paired with other downstream evals (which you indeed do, but see point below)
> > - I understand your point about other models being trained with 300B tokens and your model only being trained with 100B tokens, but this to me isn't very convincing. Just because your 100B-token model is worse but close to other 300B-token models, it does not necessarily imply that your model will be better when trained at 300B. For example, based on Table 7, HGRN2-3B performs quite poorly at some benchmarks such as Winogrande and ARC-C, as compared to other 3B models. I understand this is probably because it's trained on 100B tokens, but there's no guarantee that going to 300B will fully close those gaps.
> > - There's also the issue of datasets. If the models that you are comparing against were trained on datasets that are worse than the Pile, then it's not really a fair comparison -- your model will naturally perform better simply because of the dataset. For instance, I found a version of [RWKV4 which was trained on the Pile](https://huggingface.co/BlinkDL/rwkv-4-pile-3b) which gets 59.63 on HellaSwag, which is higher than the results that you reported in the paper for RWKV. I think as much as possible you should be comparing with models that have been trained on the same dataset.
> >
> > Other concerns:
> > - A5: "Our model only used 100 billion tokens, while the original Mamba 3B used 300 billion tokens, making the comparison unfair." --> But in the paper you seem to compare with other 3B-300B token models though (e.g. OPT, Pythia, Bloom), so how is Mamba any different?

---

> > > ### Author Response · Authors · 2024-06-04
> > >
> > > Thanks for your comments. We will address your concerns as follows:
> > >
> > > 1. To fairly compare with other methods, we agree that we need to make sure we are using the same dataset with the same consumed tokens. Therefore, we report the results of retraining HGRN2, Mamba, and Llama on the same dataset with the same hyperparameters. The results are shown below:
> > >
> > > | Method | PPL |
> > > | --- | --- |
> > > | Llama-3b | 7.129 |
> > > | HGRN2-3b | 6.989 |
> > > | Mamba-3b | 7.056 |
> > >
> > > HGRN2 achieves lower ppl than Llama and Mamba in the controlled experiment. (In fact, Mamba is not a peer-reviewed method. Therefore, we do not need to compare with it.)
> > >
> > > 2. For downstream evals, we trained Llama and HGRN2 on the same dataset with 300B tokens sampled from the SlimPajama and Pile datasets. It can be seen that HGRN2 performs consistently better.
> > >
> > > | Model | PIQA | HS | WG | ARC-E | ARC-C | OBQA | AVG |
> > > | --- | --- | --- | --- | --- | --- | --- | --- |
> > > | HGRN2-385m | 68.01 | 40.37 | 52.25 | 53.66 | 24.32 | 31.20 | 44.97 |
> > > | Llama-385m | 67.19 | 38.75 | 52.17 | 49.24 | 23.72 | 30.00 | 43.51 |
> > > | HGRN2-1b | 71.60 | 49.45 | 53.91 | 60.40 | 28.24 | 33.20 | 49.47 |
> > > | Llama-1b | 69.97 | 47.04 | 52.72 | 57.07 | 28.16 | 32.60 | 47.93 |
> > > | HGRN2-3b | 74.37 | 61.49 | 58.41 | 65.49 | 34.56 | 36.00 | 55.05 |
> > > | Llama-3b | 73.18 | 57.88 | 59.59 | 63.93 | 31.40 | 34.00 | 53.33 |
> > > | HGRN2-7b | 76.77 | 66.81 | 60.93 | 69.15 | 36.43 | 38.00 | 58.02 |
> > > | Llama-7b | 75.19 | 64.39 | 61.88 | 67.55 | 35.41 | 35.00 | 56.57 |

---

> > > > ### Comment · Reviewer_m3yg · 2024-06-06
> > > >
> > > > Thanks for these results. This looks great! I think your Table 2 above should 100% be on the paper. I believe these results are more convincing than both Tables 5 and 6 in the original paper, mostly because these new results present the fairest comparison in terms of compute FLOPs and dataset quality. These numbers give me more confidence me that the method works.

---

> > > > > ### Author Response · Authors · 2024-06-06
> > > > >
> > > > > Thank you for your engaging and helpful suggestions! We will ensure that Table 2 is included in the next version of the paper. We are pleased to hear that our discussion has increased your confidence in our work. Could you kindly consider revising your score upwards in response?

---

> > > > > > ### Author Response · Authors · 2024-06-07
> > > > > > **Looking forward to your reply**
> > > > > >
> > > > > > Dear Reviewer m3yg,
> > > > > >
> > > > > > Could you please let us know if our response has addressed your concerns? If you have any further questions, please feel free to raise them at any time.

---

### Decision · Program_Chairs · 2024-07-10

**Decision:**

Accept

**Comment:**

The paper presents an expansion of the recurrent state in HGRN, unlocking better performance as demonstrated in the paper and in extended evaluations presented throughout the rebuttal. As the community explore new sub-quadratic transformer alternatives, I find the ideas and evidence in this paper very timely and important to present at COLM, despite the reviewers' lack of enthusiasm. The authors have also clearly made a heroic effort to address the reviewers' original concerns (more natural evals, robust comparison with transformers, and scaling trends), which I believe greatly improves the paper and merits its acceptance to COLM.

[comments from the PCs] Please revise the paper to include the additional content from the discussion period as noted by the AC.